# Development of a Novel Highly Granular Hadronic Calorimeter with Scintillating Glass Tiles

**Dejing Du** [1,2,3] on behalf of the CEPC Calorimeter Working Group and **Yong Liu** [1,2,3,*] on behalf of the Scintillating Glass R&D Collaboration

1   Institute of High Energy Physics, Chinese Academy of Sciences, Yuquan Road 19B, Beijing 100049, China
2   University of Chinese Academy of Sciences, Yuquan Road 19A, Beijing 100049, China
3   State Key Laboratory of Particle Detection and Electronics, Yuquan Road 19B, Beijing 100049, China
*   Correspondence: liuyong@ihep.ac.cn; Tel.: +86-10-88236066

**Abstract:** Based on the particle-flow paradigm, a new hadronic calorimeter (HCAL) with scintillating glass tiles is proposed to address major challenges from precision measurements of jets at the future lepton colliders, such as the Circular Electron Positron Collider (CEPC). Tiles of high-density scintillating glass, with a high-energy sampling fraction, can significantly improve the hadronic energy resolution in the low-energy region (typically below 10 GeV for major jet components at Higgs factories). The hadronic energy resolution of single hadrons and the effects of key parameters of scintillating glass have been evaluated in the Geant4 full simulation, followed by the physics benchmark studies on the Higgs boson with jets in the final state. R&D efforts of scintillating glass materials are ongoing within a dedicated collaboration since 2021 with the aim to achieve a high light yield, a high density, and a low cost. Measurements have been performed for the first batches of scintillating glass samples including the light yield, emission and scintillation spectra, scintillation decay times, and cosmic responses. An optical simulation model of a single scintillating glass tile has been established to provide guidance in the development of scintillating glass. Highlights of the expected detector performance and the latest scintillating glass developments are presented in this contribution.

**Keywords:** scintillating glass; hadronic calorimeter; high granularity calorimetry; silicon photomultiplier





## 1. Introduction

High-energy electron–positron collider experiments have been proposed for precision measurements of the Higgs boson, which was discovered at the Large Hadron Collider (LHC) in 2012 [1,2], and to explore new physics beyond the Standard Model. the Circular Electron Positron Collider (CEPC), as one option among next-generation colliders as Higgs factories, requires accurate identification and reconstruction of all final states from Higgs, W, and Z bosons. Therefore, the jet energy resolution of the CEPC detector needs to achieve $\sim 30\%/\sqrt{E_{jet}(GeV)}$ [3], which poses challenges for the calorimetry system. A feasible paradigm to achieve this goal is the high granular calorimetry based on the particle flow algorithm (PFA) [4], which makes use of the optimal sub-detector accordingly to determine the energy-momentum of each particle within a jet. An essential prerequisite for calorimeters is to distinguish clusters of nearby individual particles in order to match the tracking system for charged particles and identify clusters originating from neutral particles, which can only be measured in calorimeters. PFA-oriented calorimeters with various technical options featuring high granularity to achieve an excellent three-dimensional spatial resolution are being developed and extensively studied within the CALICE collaboration [5].

As the majority of jet components at Higgs factories with a center-of-mass energy of 240 GeV are relatively low energy (mostly below 10 GeV, as shown in Figure 1), a better hadronic energy resolution would be useful for better PFA performance and jet



measurement precision. Hereby, we propose a new design for a highly granular HCAL with high-density scintillating glass tiles, with a higher-energy sampling fraction and PFA compatibility, to further improve the hadronic energy resolution. Its detector layout generally is similar to the CALICE scintillator-steel hadronic calorimetry (AHCAL) technique, proposed in the CEPC Conceptual Design Report [3], but instead of a plastic scintillator, scintillating glass tiles are instrumented.

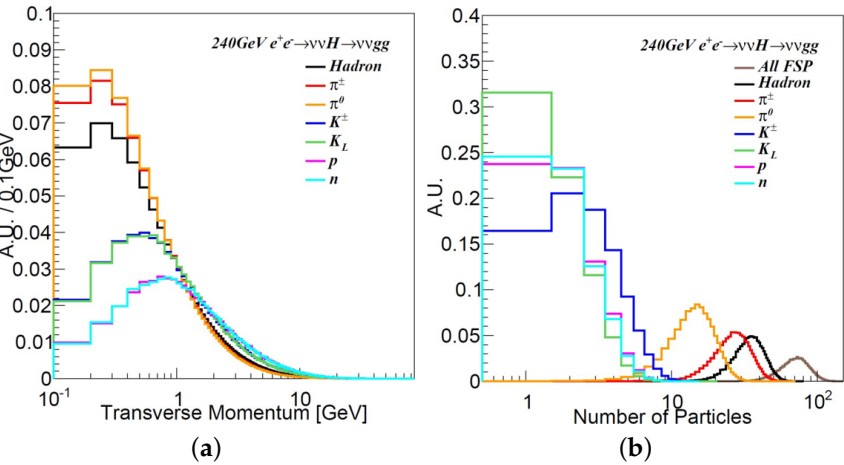

**Figure 1.** Jet components of the $e^+e^- \to \nu\nu H \to \bar{\nu}\nu gg$ at 240 GeV with distributions of (**a**) transverse momenta and (**b**) the number of different particles within a jet.

In this proceeding, Section 2 introduces the performance studies and physics potentials with this HCAL design. Recent progress of high-density scintillating glass R&D activities and characterization results of glass samples are covered in Section 3, followed by simulation studies, as well as measurements in Section 4 for an HCAL detector unit and a summary in Section 5.

## 2. Performance Studies of the Scintillating Glass HCAL

Dense scintillating glass with a moderate light yield and adjustable ingredients is usually considered as a promising option for calorimetry applications and more cost effective compared with scintillating crystals. Traditional calorimetry designs used crystals or glass in the form factor of large-volume blocks and, thus, required considerably high intrinsic light yield and transmittance. On the other hand, small-sized tiles with SiPM readout for PFA calorimetry would collect scintillation light more efficiently and, thus, significantly loosen the requirements on light yield and transmittance, which makes scintillating glass a promising option for the applications in high-granularity calorimetry.

The scintillating glass HCAL is designed as a sampling calorimeter, which consists of 40 longitudinal layers with around 4.8 $\lambda_I$ ($\lambda_I$ as the nuclear interaction length). Each layer with 0.12 $\lambda_I$ contains a steel plate as the absorber and a sensitive layer with scintillating glass tiles read out individually by silicon photomultipliers. Geant4 [6] full simulation (with version 10.7.4 and the physics list "QGSP_BERT") has been established, including all relevant physics processes for EM and hadronic showers, and its general setup is shown in Figure 2 with an HCAL module and the layer structure. Selected results on the performance evaluation and optimizations of the HCAL design are presented in the following.

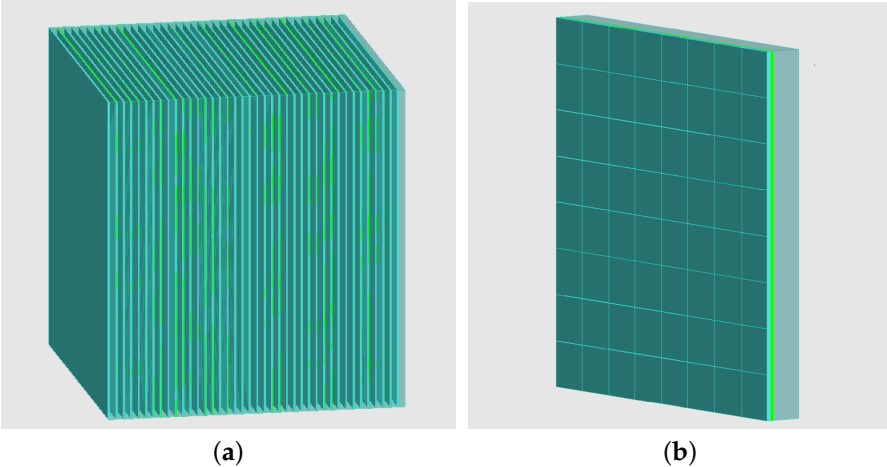

(**a**)          (**b**)

**Figure 2.** The HCAL structure visualization using Geant4: (**a**) an HCAL module with a transverse size of $108 \times 108$ mm$^2$ and 40 longitudinal layers; (**b**) a longitudinal layer including a steel plate, scintillator tiles, and 2 mm-thick readout PCB. The transverse size of the scintillator tiles is set to $3 \times 3$ mm$^2$. The thickness of a steel plate and a layer of scintillator tiles can be tuned, but with a fixed value of 0.12 $\lambda_I$ to meet the requirement of the CEPC CDR.

*2.1. Hadronic Energy Resolution*

Compared with the plastic scintillator, high-density scintillating glass can significantly increase the energy sampling fraction, which is beneficial to improve the energy resolution. Based on PFA fast simulation to factorize the jet energy resolution or the boson mass resolution, the hadronic energy resolution (obtained with single hadrons), among many key factors, ranks the second-most important factor for the PFA's performance [7].

In order to evaluate the performance potential, hadronic energy resolutions of the HCAL with plastic and glass were compared with the Geant4 simulation. The properties of the scintillating glass in the simulation corresponds to the glass sample #7 in Table 1. Three scenarios are compared and shown in Figure 3: each layer with: (1) a 3 mm-thick plastic scintillator (negligible in terms of $\lambda_I$) and a 20 mm steel plate (blue); (2) a 3 mm (0.011 $\lambda_I$)-thick scintillating glass and a 20 mm steel plate (red); (3) a 23 mm (0.084 $\lambda_I$)-thick scintillating glass (green), all with the ideal energy threshold of 0 MIP, so that all hits are effectively collected. It shows that scintillating glass HCAL is expected to have a better hadronic energy resolution especially with incident kinetic energies below 30 GeV. It should be fair to state that the plastic scintillator options provide an acceptable energy resolution, but also that scintillating glass offers substantially better performance.

For further detailed studies presented in the following, the thickness of the scintillating glass varies from 0.01 $\lambda_I$ to 0.12 $\lambda_I$, while the steel thickness is changed accordingly, so that each layer is fixed at 0.12 $\lambda_I$ in all scenarios. The $\lambda_I$ of the scintillating glass (with the constituent recipe described in Section 3) and steel are 22.4 cm and 16.8 cm, respectively. Figure 4 shows the impact of the scintillating glass thickness on the hadronic energy resolution, using neutral kaon MC samples, including two sets of energy thresholds per channel. It can be found that the thickness of the scintillating glass and the energy threshold can significantly affect the energy resolution. A lower threshold would always be desirable for better energy resolution. It should be pointed out that the glass density is another crucial factor. Figure 3 represents an earlier study with the density being 4.94 g/cm$^3$, corresponding to scintillating glass samples. In the rest of the studies, for example in Figure 3, the density used in the simulation was set to 6 g/cm$^3$, which corresponds to the goal of scintillating glass R&D.

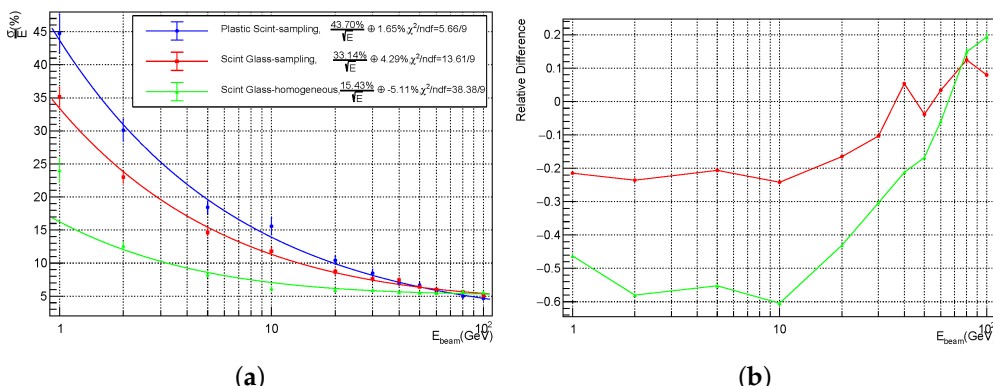

(a)　　　　　　　　　　　　　(b)

**Figure 3.** Performance with single neutral kaons ($K_L^0$) in the kinetic energy range from 1 GeV to 100 GeV perpendicular to the incidence ofthe calorimeter surface: (**a**) hadronic energy resolutions with different sensitive materials; (**b**) relative differences of hadronic energy resolutions with scintillating glass compared with with 3 mm plastic scintillator and 20 mm steel.

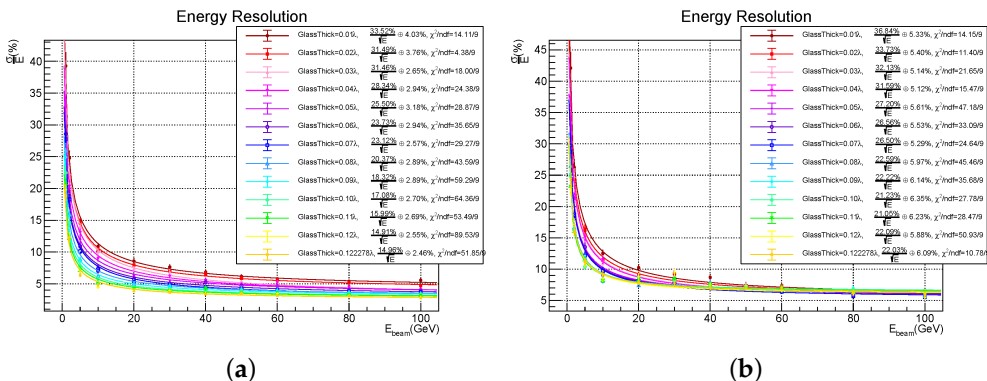

(a)　　　　　　　　　　　　　(b)

**Figure 4.** Energy resolutions of the HCAL with different thicknesses of scintillating glass for $K_L^0$ with the kinetic energy ranging from 1 GeV to 100 GeV. The glass thickness varies from 0.01 $\lambda_I$ to 0.12 $\lambda_I$ and corresponds to solid points in different colors (dark red and orange for the minimum and maximum sampling fraction, respectively) with different energy thresholds per channel of 0–0.3 MIP. The sampling fraction uses the energy deposition obtained directly from Geant4, without considering the readout implementation or corrections. (**a**) Threshold = 0 MIP per channel; (**b**) threshold = 0.3 MIP per channel.

The stochastic and constant terms of the energy resolution are extracted from each scenario in Figure 4 and shown in Figure 5. The two sets of energy thresholds of 0 and 0.3 MIP per readout channel correspond to an ideal configuration and a realistic one, respectively. Generally, with a given threshold, the stochastic term is improved with thicker glass. With the energy threshold above 0.3 MIP, the stochastic term remains almost constant when the glass becomes thicker than 0.08 $\lambda_I$. As shown in Figure 4c,d, a higher threshold significantly degrades the constant term.

It should be noted that the HCAL design is non-compensated in general, i.e., the responses to the hadronic components (denoted as *h*) and electromagnetic (EM) ones (denoted as *e*) in hadronic showers are not equal ($h/e < 1$), and normally, the $h/e$ ratio increases along with the incident particle energy [8]. Therefore, the energy resolution degrades in the high-energy region when the glass becomes thicker, as it is more sensitive to the EM components. The software compensation technique [9] in high-granularity calorimeters is a feasible option to assign different energy density weights, determined by the energy deposition per tile, to equalize responses to EM and hadronic components, which can significantly improve the energy resolution, and simulation studies are ongoing for the scintillating glass option.

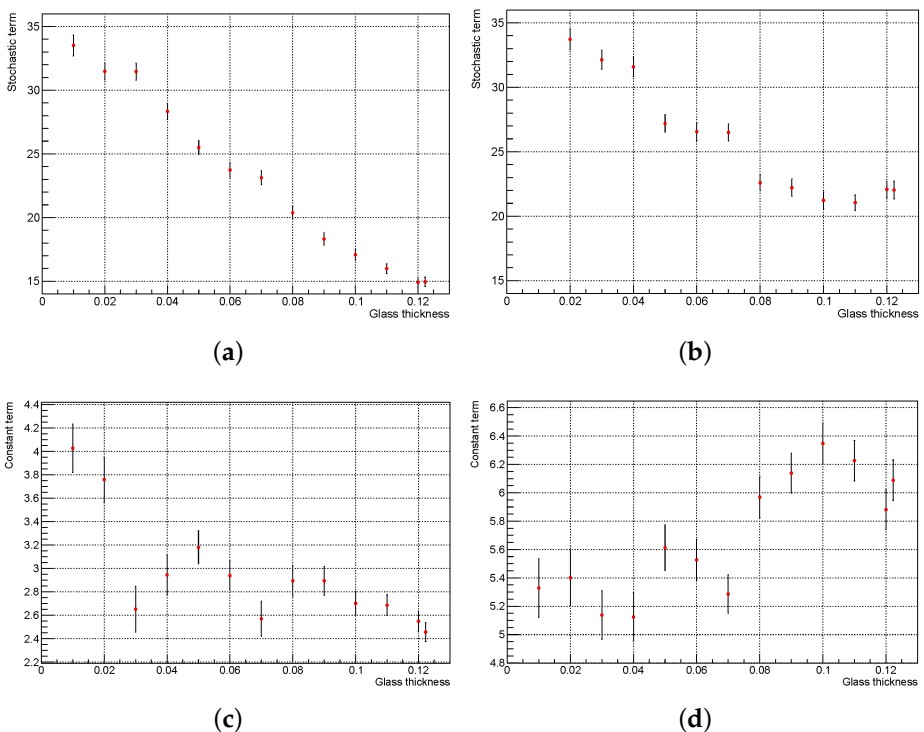

**Figure 5.** Stochastic and constant terms of hadronic energy resolution versus the thickness of scintillating glass with different energy thresholds. (**a**) Stochastic term with the threshold = 0 MIP; (**b**) stochastic term with the threshold = 0.3 MIP; (**c**) constant term with the threshold = 0 MIP; (**d**) constant term with the threshold = 0.3 MIP.

### 2.2. Boson Mass Resolution

As the majority of hadrons in jets at the CEPC are low energy, the scintillating glass HCAL has great potential for improving the jet energy resolution. Jet performance with Higgs hadronic decays has been evaluated with the scintillating glass HCAL implemented in the full CEPC detector, where the other sub-detectors are kept the same as the CEPC CDR baseline.

The boson mass resolution (BMR) is hereby used as a key parameter to quantify the physics performance. In this study of $ZH \to \nu\nu gg$ at 240 GeV, the BMR is the resolution of the Higgs invariant mass reconstructed from two gluon jets. As shown in Figure 6, the BMR with the CEPC CDR baseline detector design is around 3.8%. In the scenario of a homogeneous HCAL with scintillating glass tiles to replace the CDR baseline HCAL, the BMR is improved by around 10% to be 3.45%. A particle flow algorithm named "ArborPFA" [10] was used in the study, and the PFA-related parameters were those tuned with the CDR baseline HCAL. It is expected that the BMR can be further improved by optimizing the PFA for the scintillating glass HCAL.

Given that the stochastic energy resolution term is strongly affected by the per-channel energy threshold, it should be pointed out that a sufficiently low energy threshold (around 2.5% MIP) was implemented in the BMR simulation at this stage to illustrate the physics potentials. A range of realistic threshold values, considering possible constraints from photosensors, front-end electronics, the trigger, the DAQ, etc., will be further studied to evaluate the impacts on the BMR performance.

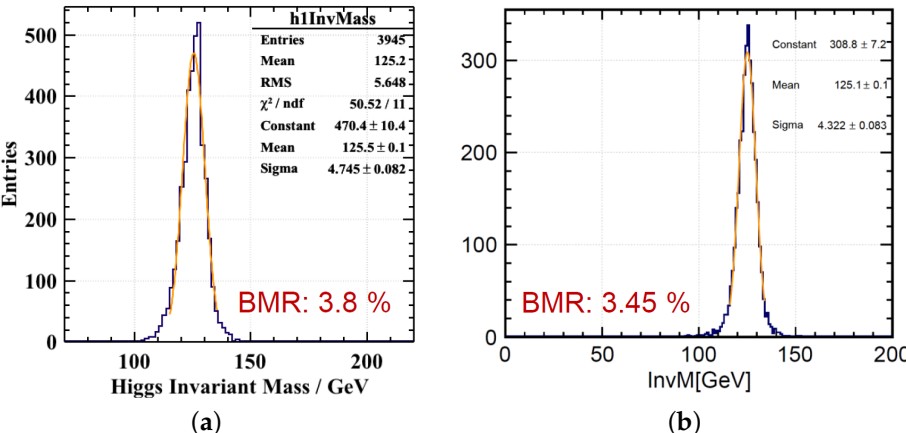

**Figure 6.** BMR of $ZH \rightarrow \nu\nu gg$ at 240 GeV with a low energy threshold around 2.5% MIP. (**a**) CEPC CDR baseline detector; (**b**) CEPC CDR baseline detector with the baseline SiW ECAL and the HCAL replaced by a homogeneous HCAL, which is 40 layers of $40 \times 40 \times 40$ mm$^3$ glass tiles (included with readout PCB and without any absorber). It needs to be noted that this setup configuration is not meant to be the final HCAL design, nor to meet the CEPC CDR requirement, but only to illustrate the physics potentials with the sufficient depth in the HCAL. Ongoing studies are being carried out with the exact same depth as the CDR requirement of around 4.8 $\lambda_I$ as the total depth.

## 3. R&D of Scintillating Glass

The R&D activities of scintillating glass materials for the CEPC-PFA-oriented hadronic calorimeter were initiated in 2021, and a scintillating glass collaboration was established in China. The collaboration aims to synthesize high-density, transparent, high-light-yield, and cost-effective glass materials and has developed several sample batches. To evaluate the glass performance, dedicated setups have been developed to measure the optical and scintillating characteristics of scintillating glass samples in the mm scale (the transverse size around $5 \times 5$ mm$^2$, the thickness around 3 mm): (1) a setup with radioactive sources (Cs-137 and Na-22) for the intrinsic light yields, energy resolutions, and decay times; (2) an ultraviolet–visible spectrometer (Lambda 650, PerkinElmer, Waltham, MA, USA) for transmission spectra; (3) X-ray sources with a tungsten target (Moxtek, MAGPRO) and a spectrometer (Omni Lambda 300i, Zolix) for X-ray excited luminescence (XEL) spectra. More details about the instrumentations and methods can be found in [11]. Over 30 pieces of samples have been measured, among which the glass sample with the best performance was aluminoborosilicate glass, with the ingredients of $B_2O_3 - SiO_2 - Al_2O_3 - Gd_2O_3 - Ce_2O_3$.

The transmittance, defined as the ratio of the light passing through to the incident light on the samples [12], is a key parameter to quantify the glass transparency and to affect its light output. The transmission spectra of scintillating glass samples are shown in Figure 7a. The absorption edge of all samples is located near 360 nm. When the wavelength is longer than 400 nm, the transmittance of sample #4 is higher than 72%. Relatively high transmittance (>75%) is required for scintillating glass.

The XEL emission spectra of seven samples (#1 to #7) are shown in Figure 7b. The measured results show that scintillating glass has broadband emissions in the range of 300 to 600 nm. The gap at around 365 nm is due to the switching of the filter in the instrument [11]. The peak of the emission spectra of all samples is around 393 nm, which matches the photon detection efficiency (PDE) spectrum of most common SiPMs.

The full-energy peak from radioactive sources is used for energy calibration. The energy resolution is defined as the FWHM (namely $2.355 \times \frac{\sigma}{mean}$) of the full-energy peak. Figure 7c shows the energy spectra of sample #7 with a $^{137}Cs$ source (662 keV gammas); its energy resolution is 27.5% at 662 keV, and the light yield was measured to be about 881 photons/MeV [11]. The intrinsic light yield of scintillating glass is aimed to be in the range of 1000–2000 photons/MeV.

Figure 7d shows the decay times of sample #7, consisting of a fast component and a slow one, which is 329 ns (20%) and 839 ns (80%), respectively. For the foreseen high-luminosity Z-pole operation at the CEPC, the decay time of scintillating glass needs to be reduced significantly and is in general required to be on the order of 100 ns.

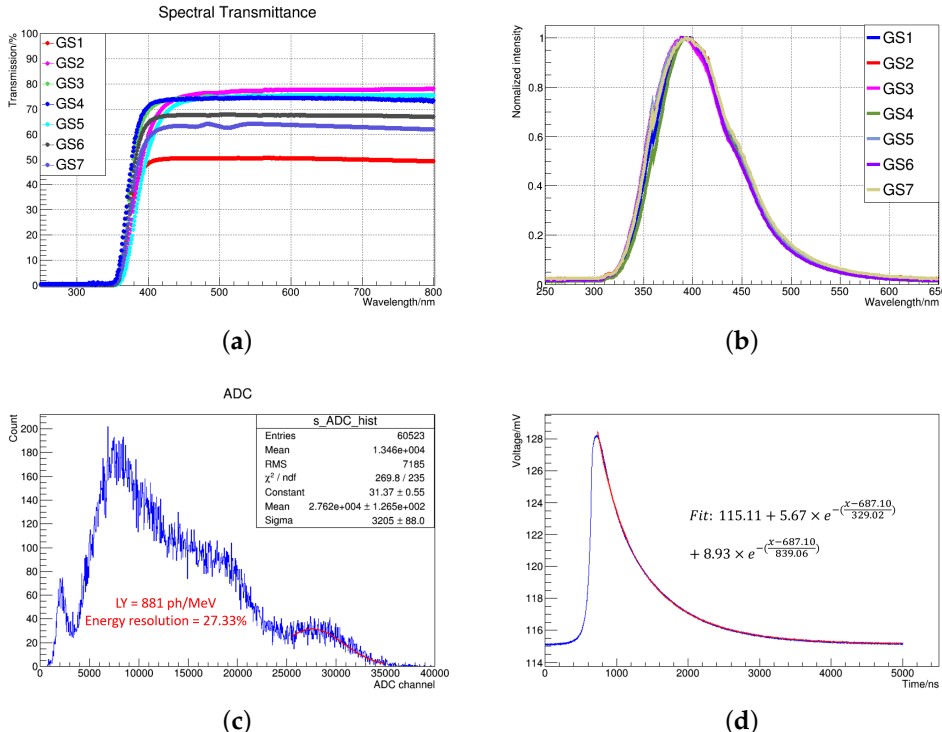

**Figure 7.** The partial measured results of aluminoborosilicate glass samples. (**a**) Transmission spectra of samples #1–#7; (**b**) normalized X-ray-excited luminescence emission spectra for samples #1–#7; (**c**) the energy spectra of sample #7 under a $^{137}Cs$ (662 keV) radioactive source; (**d**) scintillating decay time of sample #7.

Table 1 summarizes the characterization results of the optical and scintillating properties of the glass samples (all aluminoborosilicate glass). In general, glass sample #7 with a composition of $25B_2O_3 - 30SiO_2 - 10Al_2O_3 - 34Gd_2O_3 - 1Ce_2O_3$ shows the best performance, with a transmittance at visible wavelengths around 64%, a light yield of 881 photons/MeV, and a density of about 5 g/cm$^3$. The targeted values of the properties of scintillating glass are summarized as follows: a density around 6 g/cm$^3$, a light yield in the range of 1000–2000 photons/MeV, a transmittance around 75%, a decay time on the order of 100 ns. Till now, all glass samples were in the millimeter scale, and the current R&D efforts focus on the development of centimeter-scale scintillating glass samples.

**Table 1.** Characterization results of samples #1–#7 of scintillating glass.

| Sample | Density (g/cm$^3$) | Transmittance (%) | Emission Peak (nm) | Light Yield (ph/MeV) | Energy Resolution (%) | Decay Time (ns) |
|--------|--------------------|-------------------|--------------------|----------------------|-----------------------|-----------------|
| #1 | ∼4.5 | 50 | 394 | 546 | 31.04 | 273, 1007 |
| #2 | ∼4.5 | 78 | 392 | 536 | 36.47 | 334, 939 |
| #3 | ∼4.5 | 75 | 393 | 680 | 29.02 | 351, 1123 |
| #4 | 4.65 | 74 | 396 | 660 | 30.46 | 308, 1363 |
| #5 | 4.94 | 74 | 392 | 705 | 27.84 | 354, 760 |
| #6 | 4.53 | 67 | 393 | 802 | 26.77 | 318, 1380 |
| #7 | 4.94 | 64 | 394 | 881 | 27.33 | 329, 839 |

## 4. Simulation Studies and Measurements of an HCAL Detector Unit

The HCAL detector unit consists of a scintillating glass tile and a silicon photomultiplier (SiPM). Glass sample #7 ($4.5 \times 4.5 \times 3.5$ mm$^3$ after cutting and polishing) in Table 1, which was tested to have the best performance, was selected for detailed studies as the basis to extrapolate to the cm-sized tiles. The SiPM-type with the following studies was selected with the Hamamatsu S13360-6025PE [13].

### 4.1. MIP Response

The minimum ionizing particle (MIP) response of an individual detector unit provides the energy scale for the energy reconstruction of the highly granular HCAL. Muons in cosmic rays on the ground are good MIP candidates and were used for MIP calibration. Hereby, the MIP response is defined as the number of photons detected at the SiPM placed in the tile center of the transverse plane.

### 4.1.1. Cosmic Ray Experiment

As shown in Figure 8, a dedicated cosmic ray experiment was developed, using plastic scintillator tiles (as the top and bottom triggers) and the scintillating glass sample placed in between. The glass sample was wrapped with an ESR foil (with an air gap between the glass surface and the ESR foil) and directly air-coupled with a SiPM. Two trigger tiles were used for the coincidence to make sure cosmic muons pass the glass sample and constrain the incident angle range. However, as the size of the trigger tiles is larger than the sample, there was still a part of the cosmic muons with an incident angle to the glass surface normal. Figure 9a shows the MIP response of a scintillating glass tile measured by cosmic ray muons with the most probable value (MPV) of 277 detected photons at the SiPM.

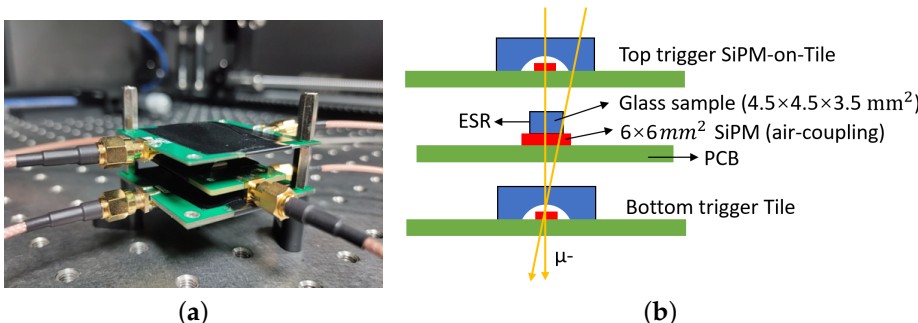

(**a**)          (**b**)

**Figure 8.** The experimental setup used for the cosmic ray test. (**a**) is the real experimental setup and (**b**) is the schematic of the experimental setup.

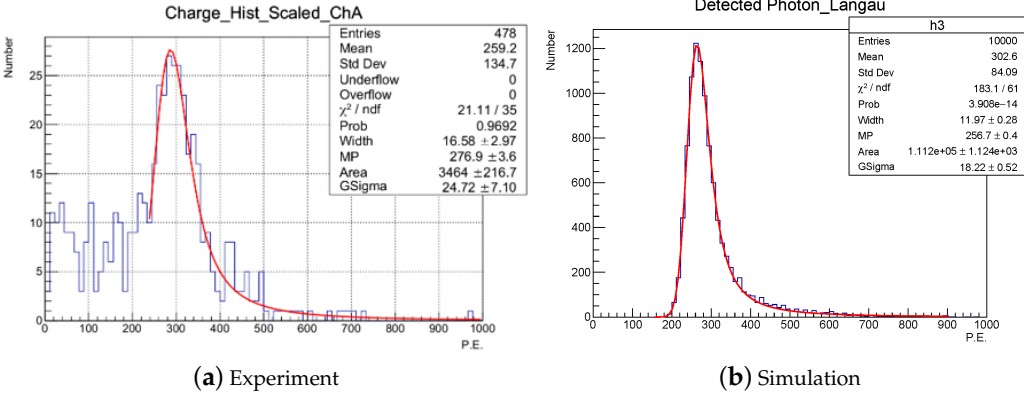

(**a**) Experiment          (**b**) Simulation

**Figure 9.** The MIP response of a scintillating glass tile: (**a**) cosmic ray test and (**b**) optical simulation.

### 4.1.2. Optical Simulation

Based on the cosmic ray experiment, the measurement data were used to validate the Geant4 optical simulation for an HCAL detector unit. As shown in Figure 10a, muons pass vertically through the tile. The scintillating glass sample contains many small bubbles, which were taken into account in the simulation, as shown in Figure 10b. Figure 9b shows the MIP response of a scintillating glass tile with the Geant4 optical simulation, with the most probable value (MPV) of 257 detected photons. This study demonstrated that the Geant4 optical simulation can well reproduce the measurements. As the muon's incidence in the simulation is exactly perpendicular to the tile surface, it is reasonable that the simulation expects a slightly smaller MIP response than the measurements.

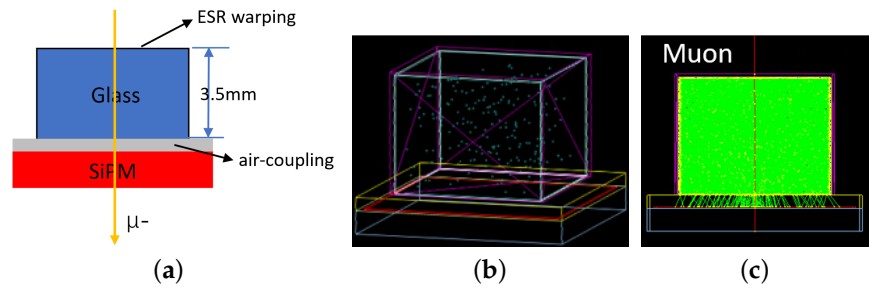

**Figure 10.** (**a**) The schematic of the geometry setup; (**b**) geometry display in the Geant4 simulation; (**c**) event display of a 1 GeV muon in the Geant4 simulation.

### 4.2. Projected Performance

At present, as the size of all samples is in the millimeter scale, the performance of the detector unit with realistic transverse size ($30 \times 30$ mm$^2$) tiles can only be obtained through simulation. Assuming that the properties of larger glass tiles remain the same as small glass samples, the response uniformity of a scintillating glass tile was studied in the Geant4 optical simulation (after validation with cosmic ray tests) by changing the incident position with a step size of 0.5 mm. The response refers to the number of photons detected at the SiPM when the incident muons pass perpendicularly through the position (X-Y in Figure 11) of a tile.

As shown in Figure 11a, the center of the tile has the highest response, as the SiPM is coupled in the tile center and has a much higher light collection efficiency. The response uniformity of the tile is represented by $\frac{max-min}{average}$, where max, min, and average are the maximum, minimum, and average values of the response. The average response and uniformity of scintillating glass tiles of different thicknesses are shown in Table 2. It can be found that the optimal thickness appears to be around 10 mm. Nevertheless, the response uniformity of the tile is far from optimal, and the uniformity needs to be improved by increasing the transmittance of the scintillating glass and optimizing the tile design.

Due to the limited transmittance of scintillation glass, a significant part o the scintillation photons will be self-absorbed by the glass. As the SiPM is located in the geometric center of the glass tile, the tile response depends on the scintillation location, determined by the position of incident particles. The farther the distance between the incident position from the SiPM, the more photons will be absorbed in the propagation process. When the incident position is within the SiPM sensitive area (hereby, $6 \times 6$ mm$^2$), the light collection efficiency is much higher, as most photons are directly detected without many reflections, leading to a much higher response in this region. When the glass becomes thicker than 3 mm, more scintillation photons are generated due to the higher energy deposition, leading to the higher response in general. On the other hand, when the glass becomes even thicker, the self-absorption effect will start to dominate and more photons will be absorbed in the glass, before being detected by the SiPM. The tendency near the tile central region with the SiPM shows that the tile response becomes lower (dimmer in in color scale) when the tile becomes thicker (e.g., thicker than 10 mm). In general, Geant4 also very much

preserves the detailed information of the optical processes and can be extracted for further quantitative studies, which would be essential for optimizing the tile design and improving the response non-uniformity.

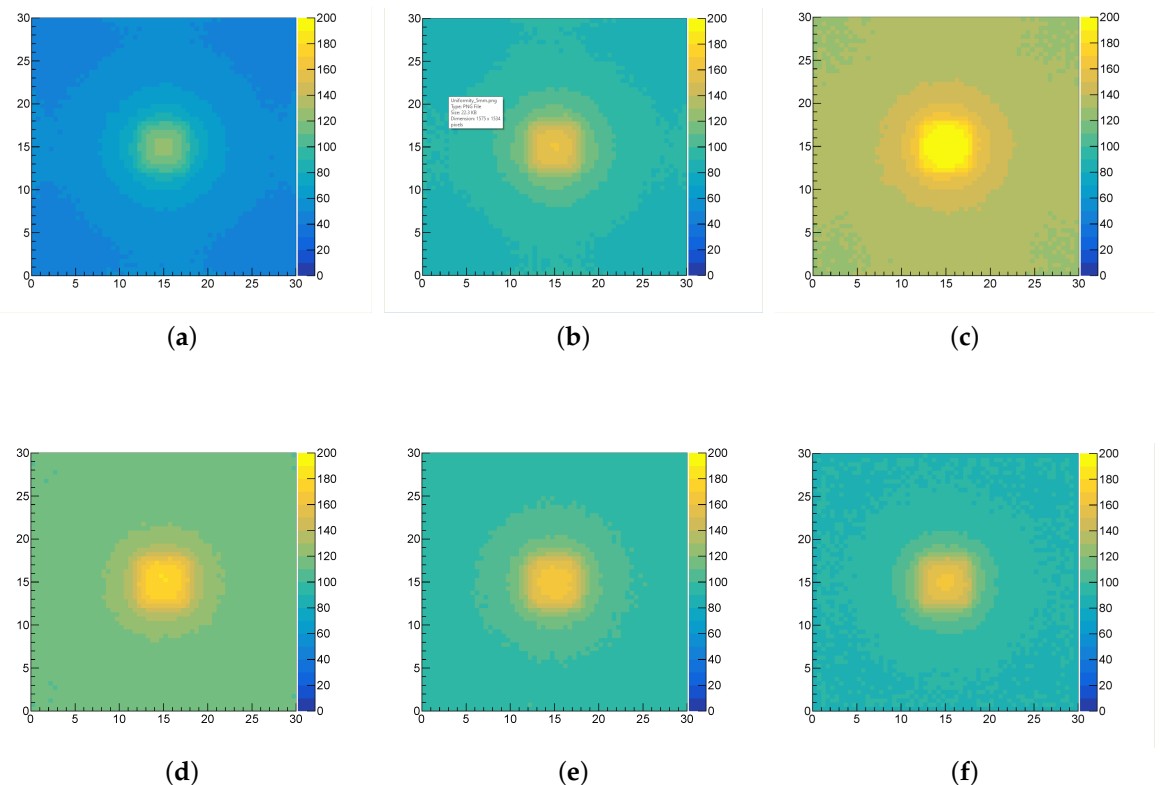

**Figure 11.** Response uniformity of different thicknesses of scintillating glass tile. Transverse size is fixed with $30 \times 30$ mm$^2$: (**a**) $30 \times 30 \times 3$ mm$^3$; (**b**) $30 \times 30 \times 5$ mm$^3$; (**c**) $30 \times 30 \times 10$ mm$^3$; (**d**) $30 \times 30 \times 15$ mm$^3$; (**e**) $30 \times 30 \times 20$ mm$^3$; (**f**) $30 \times 30 \times 25$ mm$^3$.

**Table 2.** Average response and non-uniformity with different thicknesses.

| Thickness (mm) | 3 | 5 | 10 | 15 | 20 | 23 |
|---|---|---|---|---|---|---|
| **Average Response** | 65 | 106 | 149 | 127 | 112 | 104 |
| **Non-Uniformity** | 1.1 | 0.67 | 0.47 | 0.53 | 0.63 | 0.71 |

## 5. Summary and Prospects

A new high-granularity HCAL concept with high-density scintillating glass tiles was proposed to further improve the energy resolution and the BMR. Compared with the plastic scintillator, the scintillating glass HCAL with a higher energy sampling fraction has a better hadronic energy resolution, especially with incident kinetic energies below 30 GeV. The software compensation technique [9] in high-granularity calorimeters is expected to significantly improve the energy resolution, and simulation studies are ongoing for the scintillating glass option. In addition, the scintillating glass HCAL has great potential to improve the BMR. The R&D of centimeter-scale scintillating glass with high density, transmittance, and light yield is ongoing. The cosmic ray experiment and simulation of the detector unit were carried, and the results were used as a guide for the detector design and the R&D of scintillating glass materials.

**Funding:** This project received funding support from the Chinese Academy of Sciences (CAS) Talent Program (Category B for young professionals) and the CAS Center for Excellence in Particle Physics (CCEPP).

**Data Availability Statement:** Not applicable.

**Acknowledgments:** The authors would like to express their gratitude for the fruitful discussions with many of the colleagues in the CEPC calorimeter working group and the CALICE collaboration.

**Conflicts of Interest:** The authors declare no conflict of interest.

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
