# Peer review of "Development of a Novel Highly Granular Hadronic Calorimeter with Scintillating Glass Tiles"

_instruments, doi:10.3390/instruments6030032_

Round 1

Reviewer 1 Report

This work describes a development of highly granular hadron calorimeter, optimized for studies of the Higgs boson properties with the proposed Chinese Electron Positron Collider (CEPC). The goal is to design high-performance cost-effective calorimeter, to be used with particle-flow reconstruction approach. Results from GEANT4 simulation studies on the design optimization along with research on the materials for the active media (scintillating glass) are presented.

  The article is well structured and combines results for several of the most important aspects in the development a hadron calorimeter for a specific physics program. Findings are well represented in the texts with appropriate figure/tables illustrations.

  My comments are mostly related to improvement of the clarity and details in the text and figures.

Introduction - add brief details on base-line scintillator/steel calorimeter for PFA-based CEPC, being used as reference: longitudinal and transverse granularity, interaction and radiation lengths, SiPM-on-tile approach.

L62 – add GEANT4 version and physics lists used.

Section 2.1- clarify if the sampling fraction study uses the deposited energy in the glass as reported by GEANT w/o a readout implementation or corrections considered.

L70 for the third case with 23mm thick scintillating glass setup will be useful to point out what is the total depth of the calorimeter – reduced, as the result is not compatible with fig 3a all glass case.

L91” the e/h ratio increases along with the incident particle energy.” – do you have reference or plot to illustrate/support this statement?

L93 “The software compensation” a reference is needed if particular method is being mentioned. “assign different weights” – what is being weighted?

L108 ArborPFA – needs reference.

Section 3 R&D of scintillating glass – this is important part of the development for the calorimeter – more details should be given on the glass samples – are all samples aluminoborosilicate glass, what is the difference in the composition of the 6/7 samples in the study and how it is related to the difference in the transmittance for example?

L123 it is good to mention explicitly the thickness of the samples here. Is it the 3 or 5 mm mentioned earlier in the text ?

Fig 6d – the expression for the fit should have the contributions with the same sign from the two decay time constants if I’m not missing something.

L143 What are the minimum acceptable values for the properties of the scintillating glass – decay time, light yield, transmittance, etc.

Section 4.2 What is included in the simulation as glass properties – transmittance of which sample, ESR, bubbles, etc.? What is the incident particle in this test? On fig 10 the z-axis ranges from 0 to 200? What are the units. Also, labels are overlapping between the plots in this figure...  

L184 How is the response defined?

Table 2 How do you explain the non-monotonic response behavior as function of the thickness of the tile? Why the over-the-SiPM position also manifests the same non-monotonic behavior?

Reviewer 2 Report

Summary

The article reports the current status of ongoing development and performance studies of a high-granularity hadronic calorimeter design for precision Higgs studies at future lepton colliders such as CEPC. The design has several novel aspects: it is a particle-flow-algorithm (PFA) oriented design based on small scintillating glass tiles (in contrast with the more conventional scintillator-steel sampling calorimeter in the baseline CALICE proposal) to provide a higher energy sampling fraction, light yield and resolution. Furthermore, the scintillating material for the design is a novel alumino-borosilicate glass developed specifically for the CEPC PFA-oriented calorimeter R&D program. The presented work includes Geant4 studies of energy and boson mass resolution of scintillating glass calorimeters (both sampling and homogeneous) compared with the scintillator-steel baseline, measurements and characterization results for several sample batches of alumino-borosilicate scintillating glass, a small cosmic ray experiment with one of the glass samples designed to validate the Geant4 simulation, and simulation studies of light collection and uniformity for different glass thicknesses. 

General concept comments

The article reports the status of ongoing R&D work towards developing and evaluating the conceptual design for a scintillating glass particle-flow oriented hadronic calorimeter. Glass R&D activities have produced samples with promising light yield and transmittance, early simulation studies indicate improved energy resolution over the scintillator-steel baseline design, and the optical simulation has been validated against cosmic ray data. While much work remains, I feel that the work to date and the overall conceptual design appear sound. 

I think that the article is relevant and of general interest to the field, both for the insights provided into the expected performance of scintillating-glass based particle flow calorimetry, as well as the newly-developed glass materials themselves. 

Specific comments

The work presented in the article generally appears to support the authors' conclusions that scintillating glass materials have promise for improving the performance of PFA-based hadronic calorimetry. However, the presentation of some of the methods and results have missing and/or unclear information that I would like to address: 

1. The authors state (line 61) that a "Geant4 full simulation" was established and used for the performance and optimization studies. But it is not clearly explained what a "full" simulation entails. From page 220 of the CEPC CDR (Reference 3) the active layers in the reference model consist of 3mm plastic scintillator and 2mm of readout layer (PCB). But the description of the simulation scenarios in lines 68-72 only include the glass and steel absorber layers, with no mention of the 2mm readout layer.  Please add more details about the layer structure in the simulation, ideally with an illustration of the structure. 

2. In line 25, the required jet resolution for CEPC is around 30%/sqrt(E), which (roughly) agrees with the expected jet resolution of 3-5% between 20 and 100 GeV given in the CDR. However it is hard to compare this requirement with the resolutions shown in figures 2 and 3, which are based on simulated response for single neutral kaons with, for example, a 43.7% stochastic term for the plastic-scintillator baseline design.

Similarly, the BMR estimates in figure 5 are presented without a physics requirement for comparison. 

Section 3.2 of the CDR states that "there are no yardsticks to define the requirements precisely" and that "in most cases the requirements are of the nature of the more the better". So for figures 2 and 3, is it fair to state that 43.7% corresponds  to an acceptable resolution, but that scintillating glass offers a substantially better performance? Please clarify.

Also, Section 3.2.6 in the CDR states that a BMR of 4% or better is needed to achieve a 2-sigma separation of W and Z bosons. I think this should be stated (with reference) in section 2.2

3. What is the angle of incidence for the single neutral kaons simulated in section 2.1?  Were they all normal to the calorimeter face, or over a range of angles?  Please clarify, since this is important for understanding how the particle efficiencies relate to the transverse momentum distributions. 

4. Somewhat related to the previous comment, Figures 2 and 3 show energy resolutions for single kaons over a 1-100 GeV energy range. But according to Figure 1a, the transverse momenta for single kaons and other particles are less than 1 GeV on average, and rarely exceed 10 GeV. It would be easier to understand single particle efficiencies by using  logarithmic energy scales in FIgures 2 and 3, or alternatively truncating the horizontal scale to a smaller, more relevant energy range (for instance 0 to ~20 GeV?).

5. In lines 69-71, the glass/scintillator thicknesses for the three simulations presented in Figure 2 are in units of mm, while the thicknesses in Figure 3 are given in units of lambda. To make it easier for the reader to compare Figures 2 and 3, I recommend providing lambda equivalents for the Figure 2 scenarios;  for example "3mm (0.13 lambda) of scintillating glass....", etc.  

6. Given that the stochastic energy resolution term is strongly affected by the per-channel energy threshold, did the BMR simulation studies in Section 2.2 include an energy threshold? What value?

 Please also provide more information about the simulated geometry of the "homogeneous HCAL": how many layers of glass tiles? how thick are the tiles? Are the readout layers (PCB) included in the simulation?

8. Given that glass sample #7 was selected for detailed studies for its superior combination of density, transmittance and light yield, why weren't  the measured result data for that sample included in the plots presented in Figure 6, especially those of the transmission and luminescence spectra (6a and 6b)? This feels like an odd omission to me... in fact I would have expected the sample #7 characteristics to be featured in 6c and 6d instead of #6 and #5.

9. In the cosmic ray experiment, I assume that the top and bottom "SiPM-on-tile" trigger tiles used are plastic scintillator (related to the baseline AHCAL design?), but this is not mentioned in the text.   

Reviewer 3 Report

This paper reports design and simulation for a scintillating glass-based hadronic calorimeter. While the design follows CALICE AHCAL, the use of scintillating glass is novel. A better Boson Mass Resolution of 3.45% is achieved as compared to the 3.8% CEPC CDR base-line detector. If successful, this scintillating glass-based hadronic calorimeter concept provides an interesting HCAL option for the proposed Higgs factory. The reviewer's recommendation is to publish after revisions with detailed comments listed below.

1) The authors need to explain and justify the use of "homogeneous" in Fig. 5 caption and lines 107 to describe this scintillating glass-based hadronic sampling calorimeter. The "homogeneous Hadronic Calorimeter", or HHCAL, is well defined as a hadronic calorimetry concept featured with total absorption. See, for example, the Inorganic scintillators" section of the "Particle Detectors at Accelerators" chapter in the particle data book. The use of "homogeneous" appears miss-leading to describe a sampling calorimeter.

2) The key feature of this concept is to replace organic plastic scintillator in the classical CALICE AHCAL with inorganic scintillating glass to increase the sampling fraction. The paper may benefit from a comparison between these two sensitive materials, the sampling fraction and the single hadron resolution.

3)  The paper may also benefit from a brief description on two detectors used in simulation for Fig.5, especially the CEPC CDR baseline ECAL.

4) The aluminoborosilicate scintillating glass is novel.  The paper would be clearer if the authors can provide the weight or molecular fraction of the B2O3 − SiO2 − Al2O3 − Gd2O3 − Ce2O3 glass system. It would also help if its property, such as radiation length, nuclear interaction length, dE/dx etc. can be provided.

5) Some technical details for Fig.6:

5.1) The light path length for the transmittance measurement in Fig. 6(a) should be specified. An explanation of the very diverse transmittance data would also help.

5.2)The normalization of the XEL spectra shown in Fig 6(b) needs to be specified;

5.3) The energy resolution and pedestal in Fig. 6(c) need to be defined, e.g. sigma/peak or FWHM/peak.

5.4) Instruments used to obtain all four spectra need to be defined.

6) The authors are recommended to discuss consistency between the light yield and the energy resolution listed in Table 1.

7) Ray-tracing simulation is required to understand light collection uniformity in scintillator. Such a simulation, however, is affected by many factors, such as intrinsic attenuation length and scattering centers in the glass bulk and the reflectance of the wrapping material etc. Experimental data are needed to verify these assumptions used in the simulation.
